# RADAR: Accelerate Large Language Model Inference with RL-Based Dynamic Draft Trees

## Abstract

Inference with modern Large Language Models (LLMs) is expensive and slow, and speculative sampling has emerged as an effective solution to this problem, however, the number of the calls to the draft model for generating candidate tokens in speculative sampling is a preset hyperparameter, lacking flexibility. To generate and utilize the candidate tokens more effectively, we propose RADAR, a novel speculative sampling method with RL-based dynamic draft trees. RADAR formulates the draft tree generation process as a Markov Decision Process (MDP) and employs offline reinforcement learning to train a prediction model, which enables real-time decision on the calls to the draft model, reducing redundant computations and further accelerating inference. Evaluations across three LLMs and four tasks show that RADAR achieves a speedup of 3.17x–4.82x over the auto-regressive decoding baseline.

## 1. Introduction

Modern Large Language Models (LLMs) exhibit remarkable capabilities and are widely applied across various tasks (Achiam et al., 2023; Guo et al., 2025). The parameter sizes of LLMs range from tens of billions to even hundreds of billions, and during auto-regressive generation, each generated token requires accessing entire model parameters, making LLM inference slow and costly. Here we focus on accelerating LLM inference.

To accelerate LLM inference, some methods have been proposed recently, such as prompt pruning (Zhou et al., 2023), quantization (Frantar et al., 2022), knowledge distillation (Agarwal et al., 2024), speculative sampling (Leviathan et al., 2023; Chen et al., 2023) and so on. Therein, as one of

[1]Anonymous Institution, Anonymous City, Anonymous Region, Anonymous Country. Correspondence to: Anonymous Author <anon.email@domain.com>.

Preliminary work. Under review by the International Conference on Machine Learning (ICML). Do not distribute.

the most popular methods, speculative sampling uses a small draft model to quickly generate candidate tokens, which are then verified by target LLM in parallel. Speculative sampling allows multiple tokens to be generated in a single LLM pass, significantly reducing inference latency while preserving the original output distribution of target LLM. For speculative sampling, the recent studies (Miao et al., 2024; Li et al., 2024a; Gao et al., 2025; Li et al., 2024b; Zhang et al., 2024) mostly focus on organizing and utilizing the tokens generated by the draft model more effectively.

In speculative sampling, the generated tokens are generally organized in certain data structure, termed as *draft structure* here. And there are two types of draft structures: chain and tree-based. Speculative sampling (Leviathan et al., 2023; Chen et al., 2023) first employs an auto-regressive method to invoke the draft model and produce a sequence of candidate tokens, which is referred to as a chain-structured draft. To promote parallelism during LLM's verifications, candidate tokens are organized as a verification tree (Miao et al., 2024), which is then serialized and verified via masked attention in a single LLM pass. However, most draft trees (Miao et al., 2024; Li et al., 2024a; Cai et al., 2024; Ankner et al., 2024) are static during token generation across all contexts or tasks. By reordering and pruning nodes of draft tree, methods like EAGLE-2 (Li et al., 2024b) and EAGLE-3 (Li et al., 2025) attempt to improve the acceptance length. However, the number of calls to the draft model remains a hyperparameter, lacking flexibility. For instance, we execute the source codes provided by EAGLE-3 on MT-bench (Zheng et al., 2023) and LLaMA-Instruct 3.1 8B (Dubey et al., 2024), results show that draft tokens are completely rejected at a frequency of about 31%, yet the draft model is still called 8 times(Li et al., 2025). While methods like SpecDec++ (Huang et al., 2024) and DISCO (Mamou et al., 2024) attempt to dynamically adjust the draft length by predicting the acceptance probability for each node, their methodologies are strictly designed for chain-based drafts and cannot be generalized to recent state-of-the-art tree-based methods such as EAGLE-3. Here, we attempted to dynamically decide the calls to the draft model during the draft stage, thereby generating draft trees with varying depths based on different contexts.

We propose to decide whether to invoke the draft model at

each step of draft generation process, rather than predicting the required depth of the draft tree. This is because the depth of the draft tree, equivalent to the acceptance length, is stochastic due to the rejection sampling in the speculative sampling (Leviathan et al., 2023; Chen et al., 2023), making it impossible to obtain labeled data. However, we observed that the draft tree generation process can be modeled as a Markov Decision Process (MDP), which eliminates the need for labeled data by utilizing intrinsic and extrinsic rewards. We propose to calculate these rewards based on acceptance length distributions of draft trees, which are obtained by running speculative sampling algorithm on shareGPT[1] dataset, and then all these distributions are collected to construct a dataset for training the prediction model. During training, the prediction model samples policy trajectories in real time based on the real MDP, which enables offline reinforcement learning without extrapolation error (Fujimoto et al., 2019).

Upon these insights, we propose a Reinforcement learning Adjusted Draft-generation Algorithm for speculative sampling (RADAR), leveraging the confidence scores from the draft model to dynamically decide the calls to the draft model.

In summary, our key contributions are as follows:

- We introduce a speculative sampling framework with dynamic generated draft trees, namely RADAR, which employs a lightweight prediction model to adaptively decide the calls to the draft model during the draft stage.

- To handle the lack of labeled data for training the prediction model, we model the draft tree generation process as a Markov Decision Process (MDP), and address it through a newly introduced dataset of acceptance length distribution, from which an acceptance length is sampled and then used for the calculation of action reward, eliminating the need for expensive real-time interactions with the LLM.

## 2. Related Work

### 2.1. Speculative decoding

Since the proposal of speculative decoding, researchers have been improving the algorithm from different perspectives. One line of research focuses on enhancing alignment between draft and target models via distillation(Zhou et al., 2024) or feature-level integration(Li et al., 2024a). Another direction involves optimizing draft structures, shifting from chain-based to tree-based decoding(Miao et al., 2024) and utilizing pruning and reranking(Li et al., 2024b) to

increase the acceptance rate. The third targets system parallelism to minimize drafting latency, such as Medusa's(Cai et al., 2024) parallel decoding heads and PEARL's(Liu et al., 2025) framework that overlaps the drafting and verification phases asynchronously. Our work advances the latter by introducing a dynamic draft tree that adaptively adjusts the number of calls to the draft model, resulting in minimizing drafting latency. Crucially, our approach is orthogonal to PEARL: while PEARL achieves dynamic candidate length, its primary contribution and motivation lie in maximizing temporal parallelism between draft and verification via asynchronous execution. In contrast, RADAR focus on the redundant calls to the draft model and make real-time decision to adjust the draft tree. These two paradigms can be integrated to achieve even better acceleration.

### 2.2. Dynamic draft structure

A major efficiency bottleneck in conventional speculative decoding methods stems from their inherent dependency on a fixed hyperparameter K, representing the number of the calls to the draft model. Vanilla Speculative Sampling (Leviathan et al., 2023) derives an optimal $K$ based on the i.i.d. assumption of token acceptance probabilities. To introduce context-awareness, Kangaroo (Liu et al., 2024) adopts a simple heuristic that terminates speculation if the current draft token's confidence falls below a threshold. Going a step further, SpecDec++ (Huang et al., 2024) employs a trained prediction head to estimate individual acceptance probabilities and stops speculation based on their calculated cumulative product. In a similar vein, the primary distinction in DISCO (Mamou et al., 2024) is the use of a specialized head to directly predict this cumulative product, bypassing the need for manual multiplication of individual probabilities. Despite these advancements, previous methods(Liu et al., 2024; Huang et al., 2024; Mamou et al., 2024) are inherently designed for chain-based draft, making them difficult to generalize to more complex tree-based draft structures that involve pruning and reranking(Li et al., 2024b). Furthermore, these approaches typically rely on supervised learning to estimate acceptance probabilities and use a predefined threshold to implement termination, which remains a rigid hyperparameter. In contrast, RADAR addresses these limitations by modeling the speedup ratio end-to-end using reinforcement learning. This framework enables context-aware dynamic drafting that is compatible with state-of-the-art tree-based speculative decoding methods, effectively accelerating inference across diverse contexts.

### 2.3. EAGLE Series

Distinct from vanilla speculative decoding that employs a separate small language model for token-level drafting, EAGLE (Li et al., 2024a) introduces auto-regression at the

---

[1]https://huggingface.co/datasets/anon8231489123/ShareGPT_Vicuna_unfiltered

feature level. It predicts the second-to-top layer hidden states of the target model $M_p$ and reuses $M_p$'s original LM head for sampling, which mitigates the distribution shift between the draft and target models.

The framework has evolved through two major iterations. EAGLE-2 (Li et al., 2024b) identifies that acceptance rates are context-dependent rather than purely position-dependent. It leverages the draft model's confidence scores as a proxy for acceptance probability to construct a context-aware draft tree, reordering and pruning nodes at runtime to optimize the verification efficiency. More recently, EAGLE-3 (Li et al., 2025) addresses the limitations of feature prediction and scaling. By shifting from pure feature extrapolation to direct token prediction and incorporating *multi-layer feature fusion*, EAGLE-3 significantly enhances drafting accuracy and benefits from large-scale data training.

Despite these advancements, the drafting process in the EAGLE family remains constrained by a fixed number of draft model calls. As observed in our preliminary experiments, this static hyperparameter often leads to redundant drafting efforts, highlighting a lack of flexibility in adapting to real-time verification.

## 3. Preliminaries

**Speculative Sampling** (Leviathan et al., 2023; Chen et al., 2023) has emerged as a standard technique to accelerate the inference of LLMs without compromising output quality. The core mechanism involves a collaboration between a computationally efficient draft model $M_q$ and a powerful target model $M_p$. Given an input prefix $\mathbf{x}$, the process begins with a drafting phase, where $M_q$ auto-regressively predicts $t$ future tokens. Specifically, at each drafting step $i \in \{1, \ldots, t\}$, a candidate token $\hat{x}_i$ is sampled from the distribution $q_i(\cdot) = M_q(\cdot \mid \mathbf{x}, \hat{x}_1, \ldots, \hat{x}_{i-1})$. This stage results in a draft sequence $\hat{\mathbf{X}} = \{\hat{x}_1, \ldots, \hat{x}_t\}$ along with its corresponding draft probabilities.

Following the drafting phase, the target model $M_p$ performs a single parallel forward pass on the concatenated sequence $[\mathbf{x}, \hat{x}_1, \ldots, \hat{x}_t]$ to compute the target distributions $p_1, \ldots, p_{t+1}$, where $p_i = M_p(\cdot \mid \mathbf{x}, \hat{x}_1, \ldots, \hat{x}_{i-1})$. To ensure the output follows the target distribution $p$ exactly, a speculative sampling criterion is applied sequentially. For each candidate $\hat{x}_i$, it is accepted with a probability as Eq. (1):

$$\alpha_i = \min\left(1, \frac{p_i(\hat{x}_i)}{q_i(\hat{x}_i)}\right). \tag{1}$$

If the token is accepted, the verification proceeds to $\hat{x}_{i+1}$; otherwise, the process is truncated at index $i$, and a new token is resampled from the rectified distribution as Eq. (2):

$$p_i'(x) = \frac{\max(0, p_i(x) - q_i(x))}{\sum_{x'} \max(0, p_i(x') - q_i(x'))}. \tag{2}$$

If all $t$ tokens are accepted, an additional token is sampled from $p_{t+1}$ to maximize efficiency. This ensures that in each iteration, the system generates between 1 and $t + 1$ tokens, while the final output remains statistically identical to the distribution produced by the target model alone, thereby maintaining the quality of the generation.

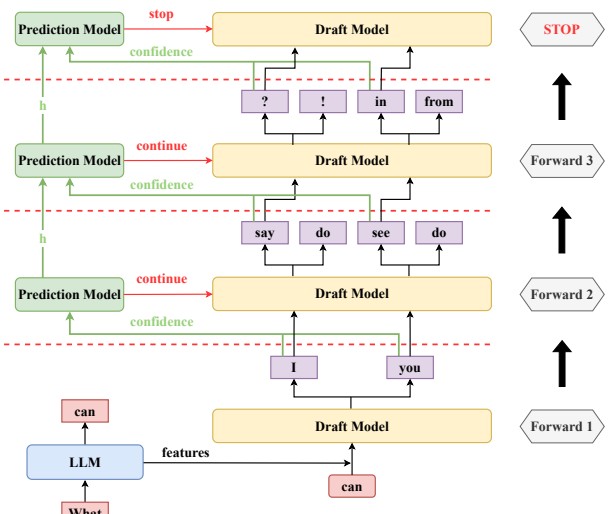

(a) RADAR

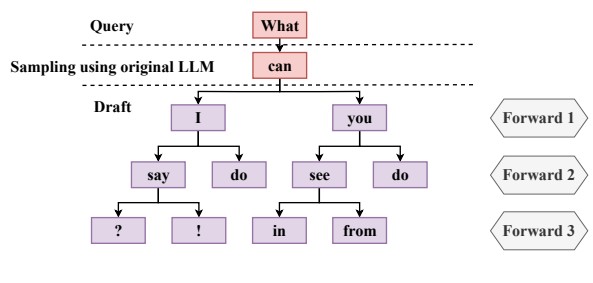

(b) Draft Tree

*Figure 1.* Pipeline of RADAR. Figure (a) illustrates the steps of predict model controlling the draft model generation, while the Figure (b) demonstrates the corresponding generation results.

## 4. RADAR

RADAR consists of three components: the target LLM, the draft model, and the prediction model, as illustrated in Figure 1 (a). Following the description of inference pipeline, we explain the core component *prediction model* in detail.

### 4.1. Inference Pipeline

Compared with previous popular speculative sampling methods such as EAGLE-3 (Li et al., 2025), we introduce a prediction model that dynamically decides to *continue/stop* the draft model's auto-regressive generation at each genera-

tion step. Here, we illustrate the inference pipeline of our approach by example, as shown in Figure 1.

First, consider "*What*" as a query into the LLM, the token "*can*" is generated. Along with some features from the LLM, the token "*can*" is then fed into the draft model for a forward pass, generating two tokens: "*I*" and "*you*", and their confidence scores are returned simultaneously. After that, the prediction model produces a control signal *continue/stop* and a hidden vector $h$ with the confidence scores as inputs. When receiving a *continue* signal, the draft model performs another forward pass with "*I*" and "*you*" as inputs, and produces four tokens: "*say*," "*do*", "*see*" and "*do*", and the top-$k$ confidence scores, corresponding to the tokens "*say*" and "*see*" in Figure 1 (a). The above steps repeat iteratively. When receiving a *stop* signal, the draft model halts further forward, and the final draft tree is generated, as shown in Figure 1 (b).

### 4.2. Prediction Model

The primary goal of the prediction model is to dynamically decide the calls to the draft model during the draft phase, enabling the construction of a dynamic draft tree. Considering the sequential nature of the generation process, our prediction model is designed as a LSTM-based network, which takes the top-$k$ confidence scores from the draft model as inputs, and outputs a binary control signal *continue/stop*, recorded as *logits*, and a hidden vector $h$.

We elaborate our prediction model on two aspects: problem modeling and model training.

**Problem modeling.** We formulate the decision of calls to the draft model as a Markov Decision Process (MDP), defined by a tuple $\mathcal{M} = (S, A, P, r, \gamma)$. The detailed definitions of these elements are as Eqs. (3-9):

- **State Space** ($S$): The state $s_t \in S$ is defined as the confidence scores of the top-$k$ candidate tokens generated by the draft model at time step $t$, as Eq. (3):

$$s_t = [c_1, c_2, \ldots, c_k], \quad 1 \le t \le t_{\max}, \quad (3)$$

where $c_i \in [0, 1]$, $k$ denotes the branching factor of the draft tree, and $t_{\max}$ is the maximum number of calls to the draft model.

- **Action Space** ($A$): The action space consists of two actions, as defined in Eq. (4):

$$A = \{0, 1\}. \quad (4)$$

Action "0" indicates the termination of draft generation, while action "1" indicates the continuation to generate candidate tokens.

- **Transition Probability** ($P$): The state transition function $P(s_{t+1}|s_t, a_t)$ is defined as Eq. (5):

$$P(s_{t+1}|s_t, a_t) = \begin{cases} \delta(s_{t+1} = s_t) & \text{if } a_t = 0 \\ P_{\text{draft}}(s_{t+1}|s_t) & \text{if } a_t = 1 \end{cases}, \quad (5)$$

where $\delta(s_{t+1} = s_t)$ represents state termination when $a_t = 0$, and $P_{\text{draft}}(\cdot)$ represents the transition probability distribution based on the draft model's generation when $a_t = 1$.

- **Reward Function** ($r$): The reward $r_t$ imposes a constant penalty $-\alpha$ at each step to discourage excessive drafting, and provides a final reward estimating the inference speed upon termination. The reward at time step $t$ is defined as Eq. (6):

$$r_t = \begin{cases} -\alpha & \text{if } t < T \\ \dfrac{\ell_{\text{acc}}}{T_{\text{gen}}(T)} & \text{if } t = T \end{cases}, \quad (6)$$

where $\ell_{\text{acc}}$ is the acceptance length. $T_{\text{gen}}(t)$ estimates the total time cost, defined as Eq. (7):

$$T_{\text{gen}}(t) = T_{\text{o}} + T_{\text{d}} \cdot t + T_{\text{pred}}(t), \quad (7)$$

$$T_{\text{pred}}(t) = \begin{cases} T_{\text{p}} \cdot (t+1) & \text{if } t < t_{\max} \\ T_{\text{p}} \cdot t & \text{if } t = t_{\max} \end{cases}, \quad (8)$$

where $T_{\text{o}}$ denotes a fix overhead, $T_{\text{d}}$ denotes the latency of a single forward pass of the draft model, $T_{\text{p}}$ denotes the latency of a single forward pass of the prediction model, and $T_{\text{pred}}$ is the inference cost of the prediction model, which is different at the maximum step $t_{\max}$, as the draft tree generation terminates, thus eliminating the need for a subsequent call to the prediction model. A breakdown of latency is provided in Appendix A.

- **Discount Factor** ($\gamma$): We set the discount factor to balance immediate and future rewards, following standard reinforcement learning practices (Mnih et al., 2015), as Eq. (9):

$$\gamma = 0.99. \quad (9)$$

**Model training.** The training loss is the negative expected reward, as Eq. (10):

$$\mathcal{L}(\theta) = -E_{\tau \sim \pi_\theta} \Big[ \sum_{t=0}^{T-1} \Big( \sum_{t'=t}^{T} \gamma^{t'-t} r_{t'} \Big) \log \pi_\theta(a_t|s_t) \Big], \quad (10)$$

where $\tau = (s_1, a_1, ..., s_T)$ is the trajectory sampled from policy $\pi_\theta$.

### 4.3. Offline Dataset

We construct an offline dataset to train the prediction model to avoid expensive real-time interactions with the LLM during training. Following the description of the dataset and its construction, we explain the use of our dataset.

For each data point $x$ in the dataset, $x = \{\mathcal{S}, \mathcal{D}\}$, where $\mathcal{S} = \{s_1, s_2, \ldots, s_{t_{\max}}\}$ denotes the sequence of states, where $s_t$ and $t_{max}$ is defined by Eq. (3); $\mathcal{D} = \{d_1, d_2, \ldots, d_{t_{max}}\}$ denotes the set of distributions for acceptance lengths corresponding to different numbers of calls to the draft model. Specifically, for each $i$, $d_i = \{p_0, p_1, \ldots, p_{t_{max}}\}$ where $p_j$ denotes the probability that the acceptance length is j given that i calls were made to the draft model.

To construct the dataset, we run EAGLE-3 on ShareGPT [1] dataset, and for each prefix in the corpus, we repeat the following procedure. First, during the draft stage, we enumerate the number of calls to the draft model from 1 to $t_{\max}$, to generate multiple draft trees. The confidence scores of the top-$k$ tokens at each generation step are collected to form the state sequence $\mathcal{S}$. Next, draft trees are serialized and verified in parallel by the LLM, and during the verification phase, for each $i$, the distribution $d_i$ of acceptance lengths for the draft tree generated with $i$ calls is derived. These distributions are then aggregated into the set $\mathcal{D}$ and we can get one data point $x = \{\mathcal{S}, \mathcal{D}\}$.

Here, we detail the derivation of the acceptance length distribution, defined by $d_i = \{p_0, p_1, \ldots, p_{t_{max}}\}$. For a draft tree node $v_1$ with parent $v_0$ and children $l_1, l_2, \ldots, l_n$, as shown in Figure 2, the probability $p_i$ used in the dataset are defined by Eqs. (11-13).

$$p_i = \sum_{v.\text{depth}=i} P_{\text{stop}}(v), \tag{11}$$

$$P_{\text{stop}}(v_1) = P_{\text{acc}}(v_0) \cdot P_{\text{acc}}(v_1|v_0) \cdot P_{rej}(l|v_1), \tag{12}$$

$$P_{rej}(l|v_1) = 1 - \sum_{j=1}^{n} P_{\text{acc}}(l_j|v_1), \tag{13}$$

where $P_{\text{stop}}(v_1)$ denotes the probability that $v_1$ is accepted while all of its children are rejected by LLM during verification stage; $P_{\text{acc}}(v_0)$ denotes the probability that node $v_0$ is accepted, which can be calculated based on speculative sampling algorithm (Leviathan et al., 2023; Chen et al., 2023); $P_{rej}(l|v_1)$ denotes the probability that all children are rejected given that their parent $v_1$ is accepted; $P_{\text{acc}}(v_1|v_0)$ denotes the probability that the child node $v_1$ is accepted given that its parent $v_0$ has been accepted, and $p_i$ denotes the probability that the acceptance length equals $i$.

To use the dataset for training prediction model, for each data point $x = \{\mathcal{S}, \mathcal{D}\}$, the state sequence $\mathcal{S}$ is input into the prediction model, which produces a complete trajectory denoted as $\tau = (s_1, a_1, \ldots, s_T)$. From this trajectory, the

number of calls to the draft model, denoted by $i$, is obtained. Subsequently, the acceptance length $\ell_{\text{acc}}$ is sampled from the distribution $d_i \in \mathcal{D}$. The reward and loss are computed using the reward function defined in Eq. (6) and the loss function defined in Eq. (10), respectively. Finally, the parameters of the prediction model are updated using policy gradient methods. Through the above steps, both the data distribution and the behavior of the predictive model in offline reinforcement learning remain consistent with those in online reinforcement learning, thereby eliminating the impact of extrapolation error on the training process (Fujimoto et al., 2019).

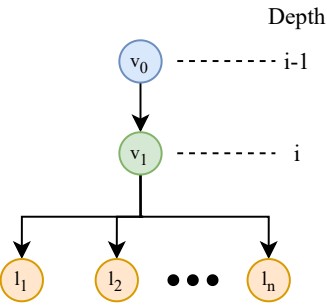

*Figure 2.* Demonstration of the node in a draft tree for the derivation of the acceptance length distribution. The node $v_1$ has parent $v_0$ and children $l_1, l_2, \ldots, l_n$, with a depth of $i$.

## 5. Experiment

### 5.1. Setup

**Target LLMs.** We conduct experiments with state-of-the-art open-source chat and reasoning models, including LLaMA-Instruct 3.1 8B (Dubey et al., 2024), Vicuna 13B (Chiang et al., 2023) and DeepSeek-R1-Distill-LLaMA 8B (Guo et al., 2025).

**Tasks.** Following EAGLE (Li et al., 2024a) and Hass (Zhang et al., 2024), we conduct evaluations on four common tasks, using the same weights for all tasks without fine-tuning on the respective tasks. For multi-turn conversation, mathematical reasoning, instruction following and code generation, we chose the MT-bench (Zheng et al., 2023), GSM8K (Cobbe et al., 2021), Alpaca (Taori et al., 2023) and MBPP (Austin et al., 2021) datasets, respectively.

**Metrics**. RADAR does not fine-tune the target LLM's weights and adopts a strict speculative sampling algorithm, ensuring no loss in performance. Thus, we do not evaluate generation quality. We use the following three metrics to assess acceleration performance:

- **Speedup Ratio:** The actual test speedup ratio relative

*Table 1.* Speedup ratios and average acceptance lengths $\tau$ of different methods setting temperature=1. L3 represents LLaMA-Instruct 3.1, V represents Vicuna and DSL represents DeepSeek-R1-Distill-LLaMA. Due to the lack of released draft model weights for DeepSeek-R1-Distill-LLaMA 8B in the original work, we could not evaluate the performance of EAGLE-2 on it. The bold text in the speedup ratio column represents the highest value under the corresponding dataset and model.

| Model | Method | MT-bench | | GSM8K | | Alpaca | | MBPP | |
|---|---|---|---|---|---|---|---|---|---|
| | | Speedup | $\tau$ | Speedup | $\tau$ | Speedup | $\tau$ | Speedup | $\tau$ |
| L3 8B | Eagle-2 | 2.56x | 3.37 | 3.43x | 3.88 | 2.89x | 4.08 | 3.29x | 4.55 |
| | Eagle-3 | 3.08x | 4.55 | 4.68x | 5.50 | 3.86x | 5.62 | 4.21x | 6.12 |
| | RADAR | **3.41x** | 4.48 | **4.82x** | 5.32 | **4.04x** | 5.51 | **4.44x** | 6.00 |
| V 13B | Eagle-2 | 2.89x | 4.38 | 3.18x | 4.46 | 2.83x | 4.09 | 3.56x | 4.93 |
| | Eagle-3 | 3.74x | 5.69 | 4.24x | 5.94 | 3.50x | 5.51 | 4.55x | 6.44 |
| | RADAR | **4.05x** | 5.67 | **4.36x** | 5.87 | **3.84x** | 5.48 | **4.75x** | 6.42 |
| DSL 8B | Eagle-3 | 3.42x | 4.88 | 4.39x | 6.39 | 3.08x | 4.49 | 3.71x | 5.35 |
| | RADAR | **3.86x** | 4.85 | **4.71x** | 6.33 | **3.17x** | 4.44 | **3.99x** | 5.31 |

to vanilla auto-regressive decoding.

- **Average Acceptance Length $\tau$:** The average number of tokens generated per drafting-verification cycle, which corresponds to the number of tokens accepted from the draft.

- **Average Number of Calls to Draft Model:** The average number of calls to the draft model per drafting-verification cycle.

The acceptance rate is not included because it only reflects the performance of the draft model. Since RADAR does not modify the structure of the draft model, the acceptance rate remains the same as that of EAGLE-3. Moreover, we introduce the average number of draft model calls because RADAR makes real-time decisions on the calls to the draft model, making it no longer a hyperparameter.

**Comparisons.** We use vanilla auto-regressive decoding as the baseline, serving as the benchmark for speedup ratios (1.00x). We compare RADAR with recent lossless speculative sampling methods, including EAGLE-2 (Li et al., 2024b) and EAGLE-3 (Li et al., 2025).

**Implementation.** Our code is built based on EAGLE-3's open-source repository. During training, we adopt the REINFORCE algorithm (Williams, 1992) for training prediction model, and set the learning rate to 1e-4. During evaluation, following EAGLE-3's setup, $k$ is set to 10, the maximum number of calls to the draft model is 8, temperature is set to 1.0 and batch size is set to 1.

### 5.2. Effectiveness of RADAR

We tested RADAR, EAGLE-2, and EAGLE-3 on four tasks: multi-turn conversation, mathematical reasoning, instruction following and code generation, across three LLMs:

LLaMA-Instruct 3.1 8B, Vicuna 13B and DeepSeek-R1-Distill-LLaMA 8B. Note that the speedup ratio is hardware-sensitive due to variations in computational capabilities, most of the previous studies (Leviathan et al., 2023; Chen et al., 2023; Miao et al., 2024; Li et al., 2024a) performed inference tests on the same environment. Thereby, in our experiments, all inference tests were run on $2\times$ NVIDIA RTX3090 GPUs. The code and weights for EAGLE-2 and EAGLE-3 are from the repository[2] in our environment.

As shown in Table 1, we present different methods' speedup ratios and acceptance lengths across four datasets, respectively. RADAR provides a speedup of approximately 3.41x–4.82x compared to vanilla auto-regressive generation, with a 3%–29% improvement over EAGLE-3. Besides, RADAR maintains a high average acceptance length, only about 1.2% lower than that of EAGLE-3. On all tested tasks and target LLMs, RADAR achieves the highest speedup ratio.

Generally, given the same model and dataset, a higher acceptance length implies a greater speedup ratio, however, RADAR deviates from this pattern. To further investigate why RADAR achieves the best speedup ratio across all tested datasets despite its average acceptance length being slightly lower than that of EAGLE-3, we examined the number of calls to the draft model in RADAR. As shown in Table 2, we present the average number of calls to the draft model per draft-verification cycle across four different datasets with three LLMs as target model, respectively. Under the configuration where the maximum number of calls is set to 8, RADAR reduces the average number of calls by 9.3%–34.3% compared to EAGLE-3's fixed 8 calls, with an average reduction of 18.7%. This effectively decreases the frequency of draft model invocations and reduces the time overhead during the draft phase.

---

[2]https://github.com/SafeAILab/EAGLE

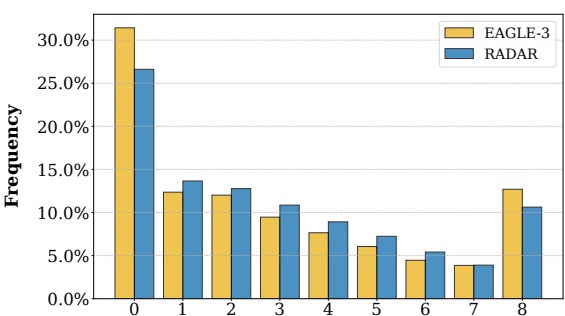

(a) The acceptance length of RADAR and EAGLE-3.

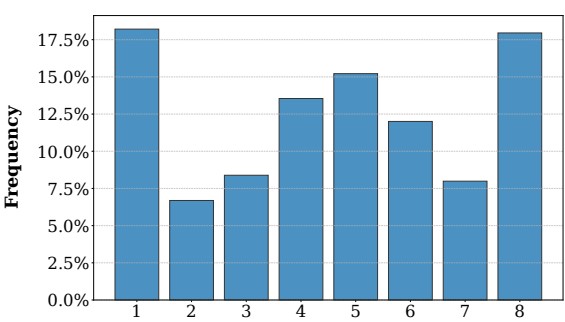

(b) The number of calls to the draft model of RADAR.

*Figure 3.* Results conducted on MT-bench and LLaMA-Instruct 3.1 8B, under temperature=1.0.

Figure 3 illustrates the distribution of acceptance length and the number of calls to the draft model on MT-bench, using LLaMA-Instruct 3.1 8B as the target model. As shown in Figure 3 (a), while maintaining a high acceptance length, RADAR significantly reduces the probability of an acceptance length of 0 compared to EAGLE-3. It also shows an increase in acceptance lengths from 1 to 7, while acceptance length 8 slightly decreases. This is because our dynamic draft tree mechanism not only reduces the calls to the draft model, but also transforms the draft tree from a tall and thin structure into a shorter and wider one, thereby increasing the acceptance probability at lower depths. Figure 3 (b) visually demonstrates how RADAR dynamically determines the number of calls to the draft model in order to create dynamic draft trees. Through the prediction model, RADAR makes early stops to the draft tree's generation adaptively when continued generation is deemed inefficient, resulting in reducing the number of calls to the draft model. This effectively decreases redundant computational overhead while maintaining almost unaffected acceptance length, further accelerating LLM inference.

### 5.3. Ablation Study

The improvement of RADAR mainly come from the prediction model, and we conducted an ablation study on MT-bench with LLaMA-Instruct 3.1 8B as the target model. The

*Table 2.* Average number of calls to draft model of RADAR under temperature=1.0. L3 represents LLaMA-Instruct 3.1, V represents Vicuna and DSL represents DeepSeek-R1-Distill-LLaMA. Following the original papers' setup, EAGLE-2 and EAGLE-3 have an preset number of calls of 8 across all models and datasets, therefore they are not included in this table.

| Model | MT-bench | GSM8K | Alpaca | MBPP |
|---|---|---|---|---|
| L3 8B | 5.25 | 6.19 | 6.20 | 6.60 |
| V 13B | 6.88 | 7.26 | 6.83 | 7.26 |
| DSL 8B | 6.10 | 7.20 | 5.85 | 6.47 |

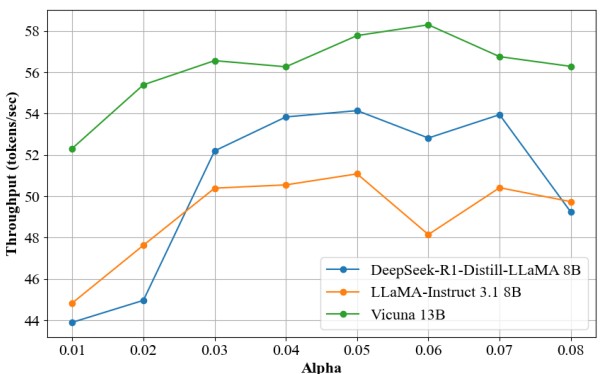

*Figure 4.* Throughput (tokens/sec) across different $\alpha$ values on MT-bench across three LLMs. The bell-shaped curves indicate an optimal trade-off between draft cost and verification efficiency.

results, shown in Table 3, indicate that prediction model in RADAR significantly enhance the speedup ratio, demonstrating the rationality of the RADAR design.

*Table 3.* Speedup ratios with LLaMA-Instruct 3.1 8B as the target model. "w/o pred" denotes the version without the prediction model.

| Method | MT-bench | GSM8K | Alpaca | MBPP |
|---|---|---|---|---|
| w/o pred | 3.08x | 4.68x | 3.86x | 4.21x |
| RADAR | 3.41x | 4.82x | 4.04x | 4.44x |

### 5.4. Robustness of $\alpha$

We analyze the sensitivity of our method to the reward penalty factor $\alpha$, a key hyperparameter in our reinforcement learning framework. We evaluate the impact of $\alpha$ on MT-bench across three models, as shown in Figure 4.

The throughput curves for all evaluated models exhibit a consistent bell-shaped trend within the range $[0.01, 0.08]$. For DeepSeek-R1-Distill-LLaMA 8B and LLaMA-Instruct 3.1 8B, the peak performance, which is 54.15 and 51.09 tokens/s respectively, is achieved at $\alpha = 0.05$. Meanwhile,

Vicuna 13B reaches its maximum throughput of 58.30 tokens/s at $\alpha = 0.06$. The consistent optimal performance around $\alpha \approx 0.05$ suggests the robustness of our reward penalty mechanism across different architectures.

Furthermore, owing to the high efficiency of our offline training pipeline, the entire training process for each model concludes within only 30 minutes. This computational efficiency allows us to easily employ a simple grid search to identify the optimal $\alpha$ for each model, ensuring the flexibility and practical deployability of our method.

## 6. Conclusion

We propose RADAR, a speculative sampling method with RL-based dynamic draft tree. We formulate the draft tree generation as a MDP and employ a prediction model to make real-time decisions of the calls to the draft model. To handle the lack in labeled data, which is challenging to obtain due to the randomness and sparsity, we ingeniously estimate the acceptance length distribution to construct a novel dataset for offline reinforcement learning in training prediction model. Experimental results on three LLMs and four tasks demonstrate that RADAR achieves a speedup of 3.17x–4.82x over vanilla auto-regressive decoding and reduce the calls to the draft model by $18.7\%$ averaged over all tested datasets, outperforming the previous methods.

Our future work will focus on improving the prediction model and training mechanisms, such as selecting models with different architectures and parameters, designing more efficient reward functions, etc. We aim to further reduce redundant calls to the draft model while maintaining a high acceptance length, thereby achieving greater acceleration for LLM inference.

## 7. Impact Statement

By eliminating redundant draft calls, RADAR contributes to the reduction of energy consumption and the operational costs of LLM deployment. We affirm our commitment to contributing positively to the society by upholding honesty and trustworthiness. All datasets used in this work are publicly available, and we ensure that our acceleration techniques are developed and maintained without incurring any form of discrimination or misuse of private data.

## 8. Acknowledgements

The authors would like to thank all the anonymous reviewers for their insightful comments.

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

## A. Breakdown of the latency

Here we provide a breakdown of the latency in Table 4. All data are collected on $2\times$ NVIDIA RTX3090 GPUs.

*Table 4.* Breakdown of the latency for $T_o$, $T_d$, and $T_p$.

| Model | $T_o$ | $T_d$ | $T_p$ |
|---|---|---|---|
| LLaMA-Instruct 3.1 8B | 0.068s | 0.0027s | 0.0004s |
| Vicuna 13B | 0.073s | 0.0025s | 0.0004s |
| DeepSeek-R1-Distill-LLaMA 8B | 0.0668s | 0.0028s | 0.0004s |

## B. Implementation Details

Our prediction model is an LSTM-based network, which consists of a single-layer LSTM followed by a two-layer MLP head. The LSTM outputs are concatenated with the original inputs and projected through the two-layer MLP to produce 2-dimensional logits, which are then transformed into a binary action distribution via Softmax. Detailed parameters of LSTM are used throughout: input_size=10, hidden_size=128, num_layers=1, dropout=0.1. Detailed parameters of MLP are as follows: hidden size=128, ReLU activation, dropout=0.1.

