# OpenReview forum: "RADAR: Accelerate Large Language Model Inference with RL-Based Dynamic Draft Trees"
_ICML.cc/2026/Conference — Submitted to ICML 2026_

### Official Review · Reviewer_T8nb · 2026-03-12

**Soundness:** 3
**Presentation:** 2
**Significance:** 2
**Originality:** 3
**Overall Recommendation:** 4
**Confidence:** 3

**Summary:**

This paper proposes RADAR, a dynamic speculative sampling framework designed to accelerate large language model (LLM) inference. The method models the generation process of the draft tree as a Markov Decision Process (MDP) and trains an LSTM-based prediction model via offline reinforcement learning. This allows for real-time decisions regarding the number of calls to the draft model, effectively reducing redundant computations.

**Compliance With Llm Reviewing Policy:**

Affirmed.

**Key Questions For Authors:**

1. If lightweight contextual features (e.g., dimensionality-reduced hidden states from the draft model's final layer, or tree depth information) were introduced into state s_t, could the accuracy of the prediction model's decisions be further improved?
2. Since the T_d parameter in the reward function strongly depends on the RTX 3090 architecture, could you provide a zero-shot inference speed degradation report (without retraining the prediction model) on different computing devices (e.g., A100/H100, or edge devices) during the Rebuttal phase?

**Limitations:**

The motivation of this paper perfectly addresses cutting-edge pain points in LLM inference optimization. The authors cleverly resolve the persistent issue of high RL training costs in speculative decoding by constructing an offline distribution dataset, representing a beautiful integration of engineering and research. The experiments at the 8B-13B scale demonstrate clear superiority over EAGLE-3. However, the oversimplified MDP state design and a severe lack of experiments on 70B+ scale LLMs prevent this paper from receiving a higher score.

**Strengths And Weaknesses:**

Strengths：
1. The authors keenly capture a core pain point in speculative decoding: fixed draft tree depths lead to significant invalid computations due to rejection sampling. Given that EAGLE-3 yields a complete rejection rate as high as 31%, exploring dynamic early exiting is a highly valuable direction from both engineering and academic perspectives.
2. To address the lack of labeled data for dynamic lengths, the paper transforms the challenge into an RL problem. More impressively, the authors accurately calculate the acceptance probability distribution of tree nodes (Eq. 11-13) to construct an offline MDP dataset. This design completely eliminates extrapolation errors and allows the training process to be completed within a mere 30 minutes, demonstrating excellent practical feasibility.
Weaknesses：
1. In the time cost estimation T_gen(t), the authors hard-coded constants T_d (draft model forward latency) and T_p (prediction model latency) specific to a particular hardware setup (2x RTX 3090). This raises a critical reproducibility and generalizability concern: if the algorithm is migrated to hardware architectures with completely different memory bandwidths (e.g., A100 or H100), the T_d value will inherently change. Will the original RL policy become entirely ineffective? The paper lacks robustness testing across different hardware architectures.
2. The primary value of speculative decoding lies in using extremely small draft models to accelerate massive target models (e.g., a 7B draft model accelerating a 70B/405B target model). However, the largest target model evaluated in this paper is only 13B (Vicuna 13B), which fails to demonstrate the required computational generational gap between the draft and target models. At this scale, the marginal effect of the prediction model's inference overhead in larger-scale systems remains unverified.

---

> ### Author Rebuttal · Authors · 2026-03-31
>
> Thank you for your feedback and recognition. We appreciate the time and effort you have taken to review our paper.
>
> > Q1: Will the RL policy become entirely ineffective on different hardware architectures?
>
> To address this concern, we conducted a cross-device ablation using LLaMA-3.1-8B on MT-bench, where RADAR-3090/RADAR-4090/RADAR-5090 denote models trained with latency parameters from the respective hardware:
>
> | Throughput (tokens/s) | on 3090 | on 4090 | on 5090 |
> |:---------------------:|:-------:|:-------:|:-------:|
> | Eagle-3               | 48.12   | 91.15   | 121.41  |
> | RADAR-3090            | **53.87**   | 99.8    | 123.02  |
> | RADAR-4090            | 47.51   | **100.4**   | 124.32  |
> | RADAR-5090            | 51.91   | 99.1    | **131.82**  |
>
> Results show that even with mismatched latency parameters, RADAR does not become ineffective. This is because the reward is composed of acceptance length and latency overhead. Acceptance length is hardware-independent and the relative magnitude ordering among $ T_o $ , $ T_d $ , and $ T_p $ remains consistent across devices.
>
> Moreover, using matched parameters achieves the best results, and since the training cost for the prediction model is only about 60 TFLOPs and 30 minutes, **device-specific tuning is trivially affordable in practice.**
>
>
> > Q2: The primary value of speculative decoding lies in using extremely small draft models to accelerate massive target models (e.g., a 7B draft model accelerating a 70B/405B target model). However, the largest target model evaluated in this paper is only 13B (Vicuna 13B), which fails to demonstrate the required computational generational gap between the draft and target models. At this scale, the marginal effect of the prediction model's inference overhead in larger-scale systems remains unverified.
>
> We address the two aspects of this concern:
>
> 1. **Prediction model overhead becomes more negligible at larger scale.** The LSTM prediction model has a fixed scale regardless of the target model size because the state, which is the concatenation of confidence scores, is independent of the hidden size. As the target LLM scales up, $T_o$ increases substantially while $T_p$ remains constant, making the prediction model's overhead even smaller in proportion.
>
> 2. **The draft-to-target model ratio in our experiments is comparable to larger-scale settings.** In our setup, LLaMA-3.1-8B uses a 0.42B draft model and Vicuna-13B uses a 0.65B draft model, which are similar to the typical 7B to 70B. Due to the time constraint and significant compute requirements, we were unable to include 70B+ experiments within the rebuttal period. We will add 70B+ results in the final version.
>
>
> > Q3: If lightweight contextual features (e.g., dimensionality-reduced hidden states from the draft model's final layer, or tree depth information) were introduced into state s_t, could the accuracy of the prediction model's decisions be further improved?
>
> We conducted an experiment using hidden states as the state representation on LLaMA-3.1-8B across four datasets, where the 4096-dimensional hidden states are projected to 128 dimensions via a linear layer before being fed into the LSTM.
>
> |  Throughput (tokens/s) | mt_bench | gsm8k | mbpp  | alpaca  |
> |:------------:|:--------:|:-----:|:-----:|:-------:|
> | Eagle-3      | 48.12    | 57.06 | 65.32 | 58.3    |
> | RADAR hidden | 48.3     | 58.6  | 65.61 | 58.14   |
> | RADAR        | **53.87**    | **59.73** | **67.78** | **60.58**   |
>
> Here RADAR-hidden denotes the variant using hidden states as input, and there is no evidence that hidden states can further improve the speedup over confidence scores. This is reasonable because **hidden states demand greater model capacity.** Using hidden states as input requires the prediction model to possess sufficient semantic understanding, which in turn necessitates more training data, longer training time, and a larger prediction model. We will include the discussion of state with hidden and tree depth in the final version.
>
> > Q4: Could you provide a zero-shot inference speed degradation report (without retraining the prediction model) on different computing devices during the Rebuttal phase?
>
> See Q1.
>
> > Q5: The MDP state design is oversimplified.
>
> Using top-k confidence scores as state is simple but effective.
>
> On one hand, confidence scores are strongly correlated with token acceptance. As noted in EAGLE-2, "there is a strong positive correlation between the draft model's confidence score and the acceptance rate of the token." Our experiments also confirm this, for example, on LLaMA-3.1-8B-Instruct and MT-bench, the Pearson correlation between the top-1 confidence score and the acceptance length is about $ r=0.4719, p=7.45e^{-137} $.
>
> On the other hand, this compact representation greatly facilitates offline dataset construction. For example, using hidden states would inflate the storage from 32 MiB to 74 GiB for EAGLE-3 on LLaMA-3.1-8B, making it impractical.

---

### Official Review · Reviewer_a493 · 2026-03-13

**Soundness:** 3
**Presentation:** 3
**Significance:** 2
**Originality:** 2
**Overall Recommendation:** 3
**Confidence:** 3

**Summary:**

This paper proposes a dynamic draft tree method for speculative decoding. It collects training data using EAGLE-3, which includes accepting or rejecting a token, and trains an LSTM-structured, hidden states-based early stop predictor.

**Compliance With Llm Reviewing Policy:**

Affirmed.

**Final Justification:**

The rebuttal has addressed most of my concerns. But as I commented, this paper lacks a valid baseline, which prevents me from understanding the effectiveness of the method. Therefore, I decide to keep my rating.

**Key Questions For Authors:**

How much data does it take to train RADAR? The entire ShareGPT dataset? How many steps/epochs does it take?

**Limitations:**

yes

**Strengths And Weaknesses:**

Strengths:
1. It identifies the fixed draft budget problem of existing models.
---
Weaknesses:
1. A fundamental problem is that RADAR has to be trained upon a trained EAGLE-3 model, which is used to collect training dataset for RADAR. It even enlarges the training time and computation cost. On the other hand, it is not clear if the trained LSTM can be generalized to other draft frameworks other than EAGLE-3.
2. Lack of experimental comparison with dynamic draft length methods, e.g., SpecDec++ and DISCO.
3. The necessity of RL formulation is questionable. There is no factual interaction between the prediction model and a real environment. So, the proposed approach is more like a learned stopping rule than a genuinely complex RL problem.

---

> ### Author Rebuttal · Authors · 2026-03-31
>
> Thank you for your time and effort in reviewing our manuscript. We sincerely appreciate your valuable comments and suggestions. Below, we provide detailed responses to your concerns.
>
> > Q1: A fundamental problem is that RADAR has to be trained upon a trained EAGLE-3 model, which is used to collect training dataset for RADAR. It even enlarges the training time and computation cost. On the other hand, it is not clear if the trained LSTM can be generalized to other draft frameworks other than EAGLE-3.
>
> **RADAR's additional training cost is negligible, and it can be generalized to other draft frameworks like HASS [1].**
>
> 1. The training cost for our prediction model is negligible: it requires only approximately 60 TFLOPs and completes within 30 minutes on our setup.
>
> 2. For generalizability, we employed our method on HASS [1] and the results are as follows.
>
>     | Speedup | MT-bench | GSM8K | Alpaca | MBPP  |
>     |:-------:|:--------:|:-----:|:------:|:-----:|
>     | HASS    | 2.49x     | 2.61x  | 2.61x   | 2.87x  |
>     | RADAR   | **2.56x**     | **2.73x**  | **2.73x**   | **3.03x**  |
>
>     RADAR consistently improves over HASS across all four benchmarks on LLaMA-3.1-8B. This is reasonable because RADAR's prediction model operates on confidence scores rather than model-specific internals.
>
> > Q2: Lack of experimental comparison with dynamic draft length methods, e.g., SpecDec++ and DISCO.
>
> We provide a comparison with DISCO [2]. Following DISCO's setup, Vicuna-13B is accelerated by Vicuna-68M and Llama-3.1-8B-Instruct by LLaMA-3.2-1B.
>
> | Model                      | Method | MT-bench | Alpaca | MBPP  | GSM8K |
> |:--------------------------:|:------:|:--------:|:------:|:-----:|:-----:|
> | Vicuna-13B-v1.3            | DISCO  | 1.46x    | 1.46x  | 1.4x  | 1.46x |
> | Vicuna-13B-v1.3            | RADAR  | **4.05x**    | **3.84x**  | **4.75x** | **4.36x** |
> | Llama-3.1-8B-Instruct | DISCO  | 1.14x    | 1.07x  | 1.23x | 1.15x |
> | Llama-3.1-8B-Instruct | RADAR  | **3.41x**    | **4.04x**  | **4.44x** | **4.82x** |
>
> RADAR substantially outperforms DISCO across all settings. Regarding SpecDec++ [3], its released weights do not cover the models used in our experiments, and due to the time constraints, we were unable to train from scratch. We will include a discussion of SpecDec++ in the final version.
>
> > Q3: The necessity of RL formulation is questionable. There is no factual interaction between the prediction model and a real environment. So, the proposed approach is more like a learned stopping rule than a genuinely complex RL problem.
>
> We appreciate this thoughtful question. The RL formulation is necessary because supervised learning is not feasible for this problem. We discussed this point in the original manuscript, Lines 56-60. Here we detail the necessity of RL formulation as follows.
>
> 1. **No deterministic labels exist.** Due to the stochastic nature of rejection sampling, the acceptance length for the same context is not a fixed value and it can be anything from 0 to 8. Consequently, the ground-truth label for a given state could be either "stop" or "continue", making it impossible to construct a consistent supervised dataset with (input, stop/continue) pairs.
>
> 2. **Supervised learning cannot yield an end-to-end control policy**. Even if one trains a supervised model to estimate the probability of stopping, this alone does not produce an optimal control policy. It lacks the ability to reason about future cumulative rewards and is unable to balance acceptance length against drafting overhead. The RL formulation handles this end-to-end by optimizing the expected return over the entire drafting trajectory.
>
> > Q4: How much data does it take to train RADAR? The entire ShareGPT dataset? How many steps/epochs does it take?
>
> We use 1,000 prompts from the ShareGPT dataset to construct the offline dataset, and train for 100 epochs with a total cost of about 60 TFLOPs, completing within 30 minutes. Only a small subset of ShareGPT is needed, not the entire dataset. We will clarify these details in the final version.
>
>
> We hope our responses have addressed your concerns. We are happy to provide further clarification if needed.
>
> References:
>
> [1]
> L. Zhang, X. Wang, Y. Huang, and R. Xu, ‘Learning Harmonized Representations for Speculative Sampling’, in International Conference on Learning Representations, 2025.
>
> [2]
> J. Mamou et al., ‘Accelerating Speculative Decoding using Dynamic Speculation Length’, arXiv preprint arXiv:2405. 04304, 2024.
>
> [3]	K. Huang, X. Guo, and M. Wang, ‘SpecDec++: Boosting Speculative Decoding via Adaptive Candidate Lengths’, arXiv preprint arXiv:2405. 19715, 2024.

---

> > ### Author Rebuttal · Reviewer_a493 · 2026-04-03
> >
> > Thank you for the responses.
> >
> > However, it still lacks comparable baselines that can be built upon the EAGLE models, since RADAR's improvement over EAGLE-3 is rather conservative. I mean applying another adaptive draft tree method on EAGLE-2/3.

---

> > > ### Author Response · Authors · 2026-04-06
> > >
> > > Thank you for the clarification. To address this concern, we applied DISCO[1] and SpecDec++[2] on top of EAGLE-3 as comparable adaptive draft tree baselines. Speedup results are as follows, where L3 8B denotes LLaMA-Instruct 3.1 8B and V 13B denotes Vicuna 13B.
> > >
> > > | Model | Method           | MT-bench | Alpaca | GSM8K | MBPP   |
> > > |-------|------------------|----------|--------|-------|-------|
> > > | L3 8B | Eagle-3          | 3.08x     | 3.86x   | 4.68x  | 4.21x   |
> > > | L3 8B | Eagle3+DISCO     | 3.10x     | 3.68x   | 4.51x  | 3.71x   |
> > > | L3 8B | Eagle3+Specdec++ | 3.12x     | 3.63x   | 4.44x  | 3.98x   |
> > > | L3 8B | RADAR            | **3.41x**     | **4.04x**   | **4.82x**  | **4.44x**   |
> > > | V 13B | Eagle-3          | 3.74x     | 3.5x    | 4.24x  | 4.55x   |
> > > | V 13B | Eagle3+DISCO     | 3.68x     | 3.47x   | 3.64x  | 4.25x   |
> > > | V 13B | Eagle3+Specdec++ | 3.60x     | 3.37x   | 3.64x  | 4.38x   |
> > > | V 13B | RADAR            | **4.05x**     | **3.84x**   | **4.36x**  | **4.75x**   |
> > >
> > > The results show that prior adaptive methods fail to consistently improve upon EAGLE-3 and often degrade performance when built on top of it. **These methods rely on supervised stopping rules that cannot end-to-end optimize the trade-off between acceptance length and drafting overhead, whereas RADAR's RL formulation directly maximizes cumulative inference speedup.**
> > >
> > > We hope this comparison addresses your concern.
> > >
> > > References:
> > >
> > > [1]
> > > J. Mamou et al., ‘Accelerating Speculative Decoding using Dynamic Speculation Length’, arXiv preprint arXiv:2405. 04304, 2024.
> > >
> > > [2]	K. Huang, X. Guo, and M. Wang, ‘SpecDec++: Boosting Speculative Decoding via Adaptive Candidate Lengths’, arXiv preprint arXiv:2405. 19715, 2024.

---

### Official Review · Reviewer_K7mX · 2026-03-13

**Soundness:** 3
**Presentation:** 3
**Significance:** 3
**Originality:** 3
**Overall Recommendation:** 5
**Confidence:** 4

**Summary:**

This paper introduces RADAR, a method that uses dynamic draft trees to accelerate LLM speculative decoding. RADAR trains a prediction model with offline RL to dynamically decide when to stop expanding the draft tree, thus improves efficiency. Experiments on three LLM families and four tasks show superior performance.

**Compliance With Llm Reviewing Policy:**

Affirmed.

**Final Justification:**

The experiments are very solid and convincing, and address my concerns.

I have raised my score to 5.

**Key Questions For Authors:**

1. Have you tried using online RL to continuously improve the prediction network? If so, comparing the reward curve would be helpful. This is just an additional question I'm curious about.

2. Why is the state modeled as the concatenation of the top-k confidence scores? Wouldn't the model hidden states be more informative?

**Limitations:**

No. Should include:

1. Reward design not tested for different devices;
2. Effectiveness on other draft tree desgin not verified

if not addressed.

---

After the authors' responses, these limitations have been addressed.

**Strengths And Weaknesses:**

- Strengths

1. The motivation is concrete. Introducing a dynamic controller to decide when to stop expanding the draft tree does help avoid redundant draft model calls.

2. Lightweight prediction model (LSTM) is efficient and easy to implement.

3. Experiments are broad, covering three LLM families and four tasks.

4. RADAR is implemented on a strong baseline EAGLE-3, while still showing consistent improvements on different settings.

- Weaknesses

1. The prediction network is trained with offline RL. The prediction network cannot continuously improve as the drafting strategy changes.

2. The reward design is complex, including not only the acceptance length, but also length penalty $\alpha$, and overhead terms depending on the system. This may challenge the robustness of the method. Although the authors conducted ablation studies on $\alpha$, the sensitivity of overhead terms is not analyzed.

3. The method is only verified on EAGLE-2/EAGLE-3, which adopts the same draft tree design. It is unclear whether the method can be applied to other tree-based drafting frameworks.

---

> ### Author Rebuttal · Authors · 2026-03-31
>
> Thank you for the thorough evaluation and insightful suggestions, which have been instrumental in refining our work.
>
> > Q1: The prediction network cannot continuously improve as the drafting strategy changes.
>
> We agree this is a valid concern in principle. However, in practice, draft models are rarely updated once deployed. Furthermore, the training cost for our prediction model is negligible, which requires only approximately 60 TFLOPs and completes within 30 minutes on our setup.
>
> > Q2: The reward design is complex and the sensitivity of overhead terms is not analyzed.
>
> The reward design intentionally mirrors the real inference latency breakdown to ensure the optimization objective is well-aligned with actual speedup. We now provide a cross-device ablation using LLaMA-3.1-8B on MT-bench, where RADAR-3090/RADAR-4090/RADAR-5090 denote models trained with latency parameters from the respective hardware:
>
> | Throughput (tokens/s) | on 3090 | on 4090 | on 5090 |
> |:---------------------:|:-------:|:-------:|:-------:|
> | Eagle-3               | 48.12   | 91.15   | 121.41  |
> | RADAR-3090            | **53.87**   | 99.8    | 123.02  |
> | RADAR-4090            | 47.51   | **100.4**   | 124.32  |
> | RADAR-5090            | 51.91   | 99.1    | **131.82**  |
>
> Results show that even with mismatched latency parameters, RADAR still works well. This is because the reward is composed of acceptance length and latency overhead. Acceptance length is hardware-independent and the relative magnitude ordering among $ T_o $ , $ T_d $ , and $ T_p $ remains consistent across devices.
>
> Moreover, using matched parameters achieves the best results, and since the training cost for the prediction model requires only about 60 TFLOPs and 30 minutes, **device-specific tuning is trivially affordable in practice.**
>
> > Q3: It is unclear whether the method can be applied to other tree-based drafting frameworks.
>
> To address this concern, we employed our method on HASS [1] and the results are as follows.
>
> | Speedup | MT-bench | GSM8K | Alpaca | MBPP  |
> |:-------:|:--------:|:-----:|:------:|:-----:|
> | HASS    | 2.49x     | 2.61x  | 2.61x   | 2.87x  |
> | RADAR   | **2.56x**     | **2.73x**  | **2.73x**   | **3.03x**  |
>
> RADAR consistently improves over HASS across all four benchmarks on LLaMA-3.1-8B. This is reasonable because RADAR's prediction model operates on confidence scores rather than model-specific internals.
>
>
> > Q4: Have you tried using online RL to continuously improve the prediction network? If so, comparing the reward curve would be helpful. This is just an additional question I'm curious about.
>
> Due to the costly real-time interaction with the target LLM required by online RL, we leave this as a promising direction for future exploration.
>
> > Q5: Why is the state modeled as the concatenation of the top-k confidence scores? Wouldn't the model hidden states be more informative?
>
> This is an insightful question. We use top-k confidence scores rather than hidden states for the following reasons:
> 1. **Confidence scores are strongly correlated with token acceptance.** As noted in EAGLE-2, "there is a strong positive correlation between the draft model's confidence score and the acceptance rate of the token." Our experiments also confirm this by calculating the Pearson correlation between the top-1 confidence score and the acceptance length, which is $ r=0.4719, p=7.45e^{-137} $ .
> 2. **Hidden states demand greater model capacity.** Using hidden states as input requires the prediction model to possess sufficient semantic understanding, which in turn necessitates more training data, longer training time, and a larger prediction model.
> 3. **Storage overhead is prohibitive.** For the offline dataset construction, for example, the hidden size of EAGLE-3 on LLaMA-3.1-8B is 4096. Storing hidden states inflates the dataset from 32 MiB (confidence scores only) to 74 GiB, making it impractical.
>
> We also conducted an experiment using hidden states as the state representation on LLaMA-3.1-8B across four datasets, where the 4096-dimensional hidden states are projected to 128 dimensions via a linear layer before being fed into the LSTM.
>
> |  Throughput (tokens/s) | MT-bench | GSM8K | MBPP  | Alpaca  |
> |:------------:|:--------:|:-----:|:-----:|:-------:|
> | Eagle-3      | 48.12    | 57.06 | 65.32 | 58.3    |
> | RADAR hidden | 48.3     | 58.6  | 65.61 | 58.14   |
> | RADAR        | **53.87**    | **59.73** | **67.78** | **60.58**   |
>
> Here RADAR-hidden denotes the variant using hidden states as input, and there is no evidence that hidden states can further improve the speedup over confidence scores.
>
> > Q6: Reward design not tested for different devices.
>
> See Q2.
>
> > Q7: Effectiveness on other draft tree design not verified.
>
> See Q3.
>
> References:
>
> [1]
> L. Zhang, X. Wang, Y. Huang, and R. Xu, ‘Learning Harmonized Representations for Speculative Sampling’, in International Conference on Learning Representations, 2025.

---

> > ### Author Rebuttal · Reviewer_K7mX · 2026-04-01
> >
> > The experiments are very solid and convincing, and address my concerns.
> >
> > I have raised my score to 5.

---

> > > ### Author Response · Authors · 2026-04-01
> > >
> > > Thanks for your acknowledgement of our work and for raising score. We appreciate your time and effort in reviewing our paper.

---

### Decision · Program_Chairs · 2026-04-30

**Decision:**

Reject

**Comment:**

This paper proposes RADAR, an LSTM-based prediction model trained via offline reinforcement learning to dynamically decide when to stop calling the draft model in tree-based speculative decoding, motivated by the observation that a substantial fraction of EAGLE-3 rounds result in full rejection.

The paper does not meet the acceptance bar due to incomplete comparative evaluation. Although the authors provided DISCO and SpecDec++ results on EAGLE-3 during the rebuttal, follow-up questions about the reproduction methodology went unanswered, leaving credibility concerns unresolved. The scalability to larger models was not demonstrated, and the justification for the RL formulation over simpler alternatives remains insufficient.